# The Role of Abstraction: Construal Level Can Improve Adolescents’ Idea Selection in the Domain of Social Creativity

**DOI:** 10.3390/jintelligence13030031

**Published:** 2025-03-05

**Authors:** Chun Li, Shuo Feng, Yue Zhang, Hui Peng, Xiaoqing Ma

**Affiliations:** 1School of Education Science, Qingdao University, Qingdao 266000, China; lichun@mails.ccnu.edu.cn (C.L.); fengshuo@qdu.edu.cn (S.F.); zhangyue65@qdu.edu.cn (Y.Z.); 2School of Psychology, Shandong Normal University, Jinan 250358, China

**Keywords:** trait-level construals, state-level construals, task domain, creative idea selection

## Abstract

Creative idea selection is an important part of the creative process, but the current research on creative idea selection has not attracted enough attention, and this lack is particularly evident in the research on creativity focusing on adolescents. In view of this, in this study, centered on the creative process, we delved into the effect of the construal level on adolescents’ creative idea selection and whether there were domain differences in this effect. The effect of trait-level construals in the general and social creative domains on creative idea selection was examined in Study 1. The results showed that (1) the creativity of the ideas selected by adolescents with high trait-level construals was higher than that of those with low trait-level construals; (2) the creativity and novelty of idea selection in the social creative domain were significantly lower than those in the general creative domain of adolescents. To determine the consistency and stability of the effects of construal level on creative idea selection, Study 2 distinguished between high and low construal levels on the basis of state initiation. It was demonstrated that (1) the creativity and applicability scores for the idea selection of adolescents with high state-level construals were higher than those of adolescents with low state-level construals; (2) the overall creativity and novelty of adolescents’ idea selection in the social creative domain were significantly lower than those in the general creative domain. On the basis of the above results, the following conclusions were drawn in this study: (1) the overall creativity and novelty of adolescents’ idea selection in the social creative domain are lower than those in the general creative domain, but there are inconsistent results with respect to applicability; (2) both high trait- and state-level construals can promote the creativity and novelty of adolescents’ social creative idea selection.

## 1. Introduction

### 1.1. Domain Generality and Specificity of Creativity

Creativity is regarded as one of the main drivers of human civilization. In everyday life, we often refer to someone as a “creative person”. The implicit assumption seems to be that a creative person possesses certain talents, traits, dispositions, and motivations that enable them to engage creatively in any activity ([6]). However, is this assumption accurate? The field of creativity has been the subject of intense and sustained debate, with researchers eager to find out “if human creativity belongs domain-general or specific to certain domains?” ([30]).

Many researchers have adopted a broad perspective, proposing that creativity is domain-general. This perspective suggests that different domains of creativity require similar components, such as divergent thinking ([58]). Therefore, a person who demonstrates creativity in one domain is also likely to exhibit creativity in other domains. A well-known example is Leonardo da Vinci. He was an Italian Renaissance painter, scientist, and inventor, excelling in painting, sculpting, and architecture while being well versed in mathematics, biology, and astronomy, establishing him as one of history’s most creative polymaths.

As research progresses, an increasing body of evidence supports the domain specificity of creativity. For instance, scholars such as [1] ([1]) have backed the notion that creativity encompasses both domain generality and domain specificity based on meta-analysis. Researchers found that educational experiences, cultural environments, and individual factors (e.g., motivation, creative personality) can predict creative performance in specific domains ([27]; [28]). This means that being creative in one domain does not imply equal creativity in others. In fact, very few individuals demonstrate outstanding creative performance across multiple domains ([5]).

Clearly, creativity cannot be simply categorized as either domain-specific or domain-general. The prevailing idea is that it is both. [31] ([31]) integrate domain specificity and generality, proposing the Amusement Park Theoretical Model of Creativity (APT). According to the APT, factors like intelligence, motivation, and divergent thinking styles ([19]) exert equal influence and predictive power on creativity across various domains. However, some factors only affect specific domains; for example, empathy is significantly associated with creativity only in the social domain ([12]).

Understanding the distinction between domain generality and specificity of creativity is crucial for educational practice. Knowing which factors generally affect creativity and which influence creativity in specific domains enables educators to more effectively address both general and specific needs ([32]).

Currently, creativity research is primarily based on assumptions of domain generalization. In the realm of special creativity research, the focus is predominantly on the scientific and artistic domains. Social creativity, as a form of creativity, occurs in everyday social interactions. Although existing research has indicated that the creativity exhibited when addressing various social issues (such as interpersonal relationships) in daily life differs from the creativity found in scientific and artistic domains ([21]; [34]), studies related to social creativity have still not received widespread attention ([61]). This study aims to explore the differences between domain-specific social creativity and general creativity.

### 1.2. The Dilemma of Creative Idea Selection Research

The creative process is an important direction in the current field of creativity research and is usually divided into two core stages: divergence (creative generation) and convergence (creative selection) ([51]). Creative generation refers to the generation of a large number of ideas with potential novelty and applicability, while creative selection requires individuals to identify and select the most creative ideas or the ideas that best meet the task goals.

Although it is well known that all successful innovations and creations are inseparable from an extremely creative good idea, merely generating a large number of ideas is not sufficient to ensure successful innovation. Against this background, numerous creativity-support techniques designed to enhance the quantity of creative ideas have come into existence ([50]). One well-known example is the brainstorming technique ([43]; [62]). The basic assumption of brainstorming is that “quantity breeds quality”, that is, generating more ideas can increase the probability of the emergence of creative ideas, and the existence of these creative ideas also increases the chances of their being selected and further developed.

However, generating more and more creative ideas does not directly guarantee the success of innovation. Successful innovation also depends on the accurate evaluation and effective selection of creative ideas. This means that a large number of ideas generated during brainstorming need to go through rigorous screening and judgment before they can be accepted and enter the subsequent development stage.

In a comprehensive review of creativity and innovation research, [3] ([3]) concluded that there is a significant disconnect between creative idea generation and implementation. Practitioners and researchers have focused on idea generation, largely neglecting the study of creative idea selection.

Why is creative idea selection overlooked? One reason may be the confidence people have in identifying and selecting the most creative ideas once they have them ([46]). However, this “confidence” is often misplaced. Theoretical ([45]; [46]; [49]) and empirical ([39]) results show that individuals perform poorly in selecting creative ideas. Individuals tend to underestimate novel ideas ([37]) or consider them inappropriate ([8]). They prefer mundane over novel ideas ([9]). Their choices are often no better than random and do not reach the average creativity level of the available ideas ([14]; [45]; [47]). Several possible reasons explain this:

Firstly, selecting creative ideas requires balancing different criteria to make a single decision. As previously mentioned, creativity has dual dimensions, and balancing novelty and applicability when selecting creative ideas is a challenge for individuals. [40] ([40]) found a negative correlation between average novelty and applicability. This suggests that it is inherently difficult to simultaneously meet both criteria, as prioritizing one standard often comes at the expense of the other.

Second, risk-related creativity research further explains this divergence in individual goals and motivations. On the one hand, individuals have a fundamental need to maintain self-integrity ([52]). In a sense, novel ideas and their associated risks to individuals and social groups can threaten an individual’s self-integrity through the uncertainty they create. When self-integrity is threatened, individuals tend to devalue or reject threatening information or its sources ([52]). On the other hand, an individual’s selection implies commitment, especially in practice. Selecting an idea means committing resources ([7]), making the stakes higher for selecting than assessing, and reinforcing existing biases.

Despite these difficulties, the ultimate goal of the creative process is to find ideas that meet both novelty and applicability criteria. Given the current dilemma in creative selection research, there is an urgent need to find effective variables to ameliorate the “underestimation of novelty” ([48]) and help individuals make more effective creative selections. Based on existing studies, this study suggests that the construal level may be a promising variable.

### 1.3. Construal Level and Creative Idea Selection

The construal level refers to how individuals mentally represent and process the abstractness and concreteness of objects ([36]). Regarding the concept of the construal level, some researchers view it as a personality trait. They argue that individuals differ in how they perceive and represent their environments: some are accustomed to abstract representations and have a high construal level, while others prefer concrete representations and have a low construal level ([18]). For example, two children tossing a ball in the yard might be described by someone with a high construal level as “the children are having fun”, while someone with a low construal level might describe it as “two little boys tossing and catching a red ball” ([56]).

However, most researchers believe that the construal level is situational, known as the state-level construals. An individual’s response depends on their mental representation of the situation, which can be either high or low. A high construal level abstracts the stimulus into invariant properties and focuses on whether the end state of the action is satisfactory. This focus is closely linked to the goal, allowing individuals to psychologically transcend the present moment. At a low construal level, individuals focus on the stimulus’s subtle details and subordinate features, forming a concrete mental representation that emphasizes the present means to achieve the desired outcome. For example, a person with a high construal level might describe going to the gym as “getting some exercise”, while a person with a low construal level might describe it as “going for a run or doing aerobics”.

Research suggests that creativity benefits from higher levels of construal due to its association with abstract thinking and broad attentional scope ([16]). High levels of construal abstract stimuli into invariant properties and focus on the end goal’s satisfaction, which is tightly tied to the goal. Currently, the greatest challenge in creative idea selection is balancing abstract long-term goals with immediate, concrete experiences to achieve optimal selection performance.

This study argues that during creative idea selection, high levels of construal may increase individuals’ focus on long-term, goal-related content, helping them recognize and favor novelty. This state encourages risk-taking and reduces focus on implementation details, allowing individuals to tap into the future potential of novel ideas.

In view of this, the present study, aiming to alleviate the underestimation of novelty, builds on previous research by examining two studies: “Does the level of construal positively influence an individual’s selection of creative ideas?” and “Does this effect differ by domain (general and social creativity)?”. This study will deepen the research into creativity and the creative process, providing empirical evidence for the development of creativity.

## 2. Preliminary Study

A preliminary study was conducted to collect ideas for use as research material in the formal study. The preliminary study randomly recruited 128 adolescent subjects (64 male and 64 female high school students; 64 freshmen and 64 sophomores) via posters in China. All participants had normal or corrected vision, typed at least 20 words per minute, had no prior experience with similar experiments, and signed an informed consent form before the study.

The preliminary study took place in a computerized room where each subject received a printed copy of the research task description and brainstorming rules ([42]). The AUT task (general creativity domain) developed by [25] ([25]) and the social creativity task (social creativity domain) by [24] ([24]) were selected as the research tasks (see Study 1 for details). This study used a between-groups design, randomly assigning participants to one of two conditions. They opened an electronic document, generated as many creative ideas as possible according to the task requirements, and saved their work for 15 min. Participation was anonymous to reduce assessment anxiety.

Subsequently, two master’s students familiar with creativity research carefully read and examined all the original ideas, removing irrelevant and repetitive ones and integrating incomplete ideas. The researcher compiled a scoring manual based on the study’s purpose and content. Four experienced raters were trained on the definitions of creativity, novelty, applicability, and the main scoring rules and points to note.

Among them, the creativity of an idea is the degree to which it is novel and applicable ([4]). Creativity of all ideas is rated on a 5-point scale (1 = very uncreative–5 = very creative). Novelty is the degree to which the idea is new; a novel idea is uncommon and proposed by a few people ([11]). The novelty of all ideas was rated on a 5-point scale (1 = very not novel–5 = very novel). Applicability refers to the feasibility of the idea and the effectiveness of the solution to the problem ([11]), and the applicability of all ideas was rated on a 5-point scale (1 = very not applicable–5 = very applicable). At the end of the training, four raters were randomly divided into two groups of two and the creativity, novelty, and applicability of each point of the idea were rated.

The total valid ideas obtained were 591 for the general creativity domain task (AUT) and 207 for the social creativity domain task (emotional distress). In this study, the internal consistency coefficients of the raters on creativity, novelty, and applicability were calculated using the Intraclass Correlation Coefficient (ICC) to determine the reliability of the ratings. The ICC for the AUT task was 0.966 for creativity, 0.945 for novelty, and 0.967 for applicability. The ICC for the emotional distress task was 0.975 for creativity, 0.841 for novelty, and 0.920 for applicability. The scoring results of the two tasks reached a high degree of consistency in terms of creativity, novelty, and applicability, so the mean creativity, novelty, and applicability scores of the two raters were calculated separately as the creativity index for each idea.

The selected ideas were chosen based on their comprehensive creativity scores from the pilot study, which ensured a balanced representation of ideas across different creativity levels. From each of the five score ranges (1–5), two ideas were selected, forming two pools of ten ideas each, as shown in Figure 1 and in Figure 2 below:

## 3. Study 1

In previous studies, the construal level refers to an individual’s degree of abstraction in information processing, categorized into trait-level construals and state-level construals. Trait-level construals are usually measured using tools like the Behavior Identification Form ([60]), reflecting an individual’s stable cognitive tendencies. In contrast, state-level construals are manipulated through research tasks (e.g., How and Why? tasks, holistic vs. local processing tasks, etc.) ([63]; [20]).

High levels of trait construal are often associated with abstract thinking and long-term goal pursuit, which may support creative idea selection. However, there is currently no empirical research directly examining the specific impact of high trait-level construals on performance in creative tasks. Furthermore, existing studies exploring the relationship between construal levels and creativity have primarily focused on general domains of creativity (e.g., [38]). In contrast, tasks in the social creativity domain, such as social problem-solving or teamwork scenarios, typically require individuals to mobilize more specific cognitive and emotional resources ([2]). This characteristic distinguishes it from general domains of creativity; in other words, the social creativity domain may be influenced by more inhibitory factors during the development of creativity ([23]), resulting in varying intensities of the effects of trait-level construals across different types of creative tasks. Based on this, we propose the following hypothesis:

**H1a:** 
*Adolescents with high levels of trait construal outperform those with low levels of trait construal in creativity tasks;*


**H1b:** 
*Adolescents with high levels of trait construal perform better on creativity in general creativity domain tasks than in social creativity domain tasks.*


### 3.1. Study 1 Method

#### 3.1.1. Participants and Design

A total of 130 adolescents participated in Study 1 (76 boys and 54 girls), with an age range of 16–18 years old, and all signed the informed consent form. The Chi-square test results showed that the distribution of different age groups was uniform (χ^2^ = 0.431, *df* = 2, *p* = 0.806), indicating that the random assignment was effective at the age level and that the age distribution would not have a significant impact on the experimental results.

This study assessed participants’ trait-level construals using the Behavioral Identification Form (BIF). Referring to the scoring criteria of [26] ([26]), participants scoring ≥8 on the BIF were categorized into the high trait-level construals group (65 participants), and those scoring <8 were categorized into the low trait-level construals group (65 participants).

An independent samples *t*-test showed that the participants in the high trait-level construals group had significantly higher BIF scores than those in the low trait-level construals group (*t* = 18.15, *p* < 0.001, *d* = 3.19). The mean BIF score for the high trait-level construals group was significantly higher than that of the low trait-level construals group (*M*
_high_ = 9.12, *M* _low_ = 5.80), indicating a reasonable and effective grouping method.

#### 3.1.2. Apparatus and Stimuli

Behavioral Identification Questionnaire: The BIF ([59]) is a trait-based questionnaire for characterizing the degree of abstraction of a behavior. The original questionnaire consists of 25 items, each describing a behavior with two explanations. Participants select the construal that seems most relevant. Drawing on the work of [57] ([57]), this study utilized a shortened version of the BIF, consisting of 10 items randomly selected from the original set of 25. Participants were asked to choose the description of the behavior that best matched their preferences. According to the scoring method proposed by [26] ([26]), selecting an abstract construal was assigned a score of 1, while a concrete construal received a score of 0. The total score for the 10 items, ranging from 0 to 10, was then calculated. Using the median split method ([33]), a BIF score of ≥8 was categorized as high trait-level construals, while a score of <8 was classified as low trait-level construals.

Materials of Creative Idea Selection

General Creative Domain Task “AUT”: Is there an unusual use for the everyday object of a paper cup other than to hold water? ([25]).

Social Creative Domain Task “Emotional Distress”: “Noonan is having trouble controlling Ta’s recent emotions. Ta feels unhappy and plagued by an indescribable loss of control. What should Ta do?” ([24]).

Manipulation of Test Material

To ensure the research manipulation met expectations ([29]), subjects answered the question “I chose the idea that I thought was the most creative” (1 = completely disagree–7 = completely agree).

Apparatus

The equipment required for this study included desktop computers and electronic files. All computers had partitions between them, and the mouse and keyboard were silent devices to ensure that the subjects’ research environment was relatively independent, eliminating social comparison or convenience interference.

#### 3.1.3. Procedure

This study divided participants into a high trait-level construals group (65 participants) and a low trait-level construals group (65 participants) based on their BIF scores, with both groups participating simultaneously. The flow of this research is shown in Figure 3 below.

First, the researcher introduced the research procedure and criteria for selecting creative ideas (novel and applicable). Participants were instructed to complete the task based on their intuition ([13]; [44]), which lasted 3 min.

Next, participants completed idea selections for the general creative domain task (AUT) and the social creative domain task (emotional distress), as shown in Figure 4. Each task consisted of 45 pairs of ideas (generated by permutations of 10 ideas from the preliminary study), presented randomly on either side of the screen (left or right). Each pair was presented for 10 s. The participants selected the most creative idea by pressing a key: “F” for the left side and “J” for the right side. The total time for the idea selection task was 10 min.

Finally, the subjects completed the manipulation test, which lasted 10 min.

Data analysis was performed using Excel to organize the raw data, followed by an independent samples *t*-test and ANOVA with SPSS 27.0.

### 3.2. Study 1 Results

For Study 1, the performance of adolescents with high and low levels of trait construal in the general creative domain and social creative domain tasks is presented in Table 1. To test the interaction effect of trait-level construals and task domain on adolescents’ creative idea selection, three 2 (high trait-level construals, low trait-level construals) × 2 (general creativity domain task, social creativity domain task) repeated-measures ANOVAs were conducted, with creativity, novelty, and applicability as the dependent variables, trait-level construals as the between-group variable, and task domain as the within-group variable.

For creativity, there was a significant main effect of trait-level construals, *F* (1, 128) = 5.74, *p* < 0.05, *η*^2^*_p_* = 0.04. The creativity scores for idea selection were significantly higher for subjects with high trait-level construals than for those with low trait-level construals. There was a significant main effect of task domain, *F* (1, 128) = 4.94, *p* < 0.05, *η*^2^*_p_* = 0.04, with creativity scores for idea selection in the general creativity domain task significantly higher than in the social creativity domain task. The interaction effect between trait-level construals and task domain was not significant, *F* (1, 128) = 2.15, *p* > 0.05, *η*^2^*_p_* = 0.02.

For novelty, the main effect of trait-level construals was not significant, *F* (1, 128) = 1.34, *p* > 0.05, *η*^2^*_p_* = 0.01. The main effect of the task domain was significant, *F* (1, 128) = 32.03, *p* < 0.001, *η*^2^*_p_* = 0.20, with novelty scores for idea selection in the general creativity domain task significantly higher than in the social creativity domain task. The interaction effect of trait-level construals with the task domain was not significant, *F* (1, 128) = 0.34, *p* > 0.05, *η*^2^*_p_* = 0.003.

For applicability, the main effect of trait-level construals was not significant, *F* (1, 128) = 1.92, *p* > 0.05, *η*^2^*_p_* = 0.02. The main effect of the task domain was not significant, *F* (1, 128) = 1.75, *p* > 0.05, *η*^2^*_p_* = 0.02. The interaction effect of trait-level construals with the task domain was also not significant, *F* (1, 128) = 2.01, *p* > 0.05, *η*^2^*_p_* = 0.02.

According to the results of Study 1, there was a significant main effect of trait-level construals on adolescents’ creative idea selection. Adolescents with high levels of trait construal scored significantly higher in terms of creativity than those with low levels of trait construal in both the general and social creativity domain tasks. This indicates that adolescents with high levels of trait construal showed higher creativity in the studies, thus validating hypothesis H1a.

The main effect of the task domain on creativity and novelty is significant. In general creative domain tasks, adolescents’ scores on the creativity and novelty of creative idea selection are significantly higher than those in social creative domain tasks, which supports the task domain differences proposed in hypothesis H1b. However, the interaction effect between the trait-level construals and the task domains does not reach a significant level, indicating that H1b is only partially supported.

### 3.3. Study 1 Discussion

Study 1 compared, for the first time, differences in performance between adolescents with different levels of trait construal on general and social domain tasks of creative idea selection. Overall, the findings support the influence of trait-level construals on adolescents’ creative idea selection. However, differences in task domains and results for novelty and applicability did not show consistent effects. These findings provide new perspectives on the effects of trait-level construals on creative idea selection and suggest complex mechanisms that may need further exploration.

#### 3.3.1. The Effect of Trait-Level Construals on Creative Idea Selection

In terms of creativity in idea selection, the results showed that adolescents with high trait-level construals scored significantly higher than those with low trait-level construals, consistent with previous research ([10]). Individuals with high construal levels can understand tasks from a more abstract and long-term perspective, a trait that helps them show higher levels of creativity in the face of uncertainty and complexity. This finding re-emphasizes the role of construal level in creative tasks, supporting its role as a key cognitive disposition that drives innovative thinking.

This result can also be explained by the Construal Level Theory (CLT, [54]). Based on this theory, adolescents with high levels of trait construal tend to adopt a more abstract way of thinking. This abstraction-oriented cognitive model helps them deal with uncertainty and ambiguous information, leading to more creative selection in creative tasks. In contrast, individuals with low levels of trait construal are more concerned with concrete and immediate details. They may tend to select direct and concrete solutions in creative tasks, lacking sensitivity to novelty and limiting creativity.

In addition, adolescents with high-level construals may perform better on intelligence tests, and this general intelligence may be more effective in accurately evaluating the creativity of ideas ([53]).

#### 3.3.2. The Effect of Task Domain on Creative Idea Selection

The main effects of the task domain were also validated. Compared to the general creativity domain task, adolescents’ creativity and novelty scores were significantly lower in the social creativity domain task. This is consistent with previous research. [23] ([23]) argued that social creativity tasks are more complex than general creativity tasks. Individuals are more influenced by subtle social factors, such as social norms, interpersonal atmosphere, and interaction pressure, which inhibit their social creativity performance. In fact, in social creative tasks, the performance range of creativity and novelty is restricted, which makes the evaluation task more difficult. In addition, since the tasks in the social creative domain are carried out after the AUT tasks, to a certain extent, they may be affected by the fatigue effect and decreased attention of the subjects.

The difference in performance on the novelty and applicability dimensions suggests that the openness and abstraction of tasks in the general creativity domain provide adolescents with a broader space for thinking. However, performance on the applicability dimension was more complex for social creativity domain tasks. This may be because social domain tasks require consideration of both practical feasibility and social acceptance of ideas, resulting in limited novelty but relatively balanced applicability scores. This echoes [15] ([15]) theory on the role of psychological distance in novelty facilitation. Novelty is related to hypotheticality (something novel, like something hypothetical, has not yet been experienced), making it distant in psychological terms, which facilitates abstract processing. It also suggests that in the applicability dimension, the social context’s focus on practical application value may offset some advantages of open-ended tasks.

#### 3.3.3. The Effects of Trait-Level Construals and Task Domains on Creative Idea Selection

According to Study 1, adolescents exhibited superior performance in the general creativity domain compared to the social creativity domain, although the interaction effect between trait-level construals and the task domain did not reach a significant level. Especially in terms of creativity and novelty dimensions, the performance in general creative domain tasks is more prominent; however, in the applicability dimension, tasks in the social creativity domain, which are more aligned with actual needs, score relatively higher than those in the general creativity domain. These results suggest a stable and consistent predictive effect of task domains on adolescents’ creative idea selection, indicating that the practical application of creativity may be more influenced by social contexts.

## 4. Study 2

The trait-level construals measured in Study 1 preliminarily show that adolescents with a high level of trait construal have an advantage in creative idea selection. To determine the stability of the influence of construal level, Study 2 changed the manipulation of construal level ([57]) and tested the influence of state-level construals and task domain on adolescents’ creative idea selection through research manipulation. We propose the following hypothesis:

**H2a:** 
*Adolescents with high levels of state construal outperform adolescents with low levels of state construal on a creative task;*


**H2b:** 
*Adolescents with high levels of state construal show higher levels of creativity performance in general creativity domain tasks than in social creativity domain tasks.*


### 4.1. Study 2 Method

#### 4.1.1. Participants and Design

We randomly re-recruited 129 adolescents for Study 2 (52 boys, 77 girls), aged 16 to 18, all of whom provided informed consent. The Chi-square test results indicated that the age group distribution was uniform (χ^2^ = 0.140, *df* = 2, *p* = 0.933), demonstrating that random assignment was effective at the age level, and that age distribution would not significantly impact the study’s results.

#### 4.1.2. Apparatus and Stimuli

The research followed the CLT research paradigm ([17]), using the How and Why startup task to manipulate the state-level construals, as shown in Figure 5. Subjects with low state-level construals were asked to think about and fill in “How to maintain a good mood”, while subjects with high state-level construals were asked to think about and fill in “Why to maintain a good mood”.

Materials of Creative Idea Selection. Same as Study 1.

Manipulation of Test Material

To ensure that the research manipulation meets expectations ([29]), this study included a manipulation test. For the manipulation of creative idea selection, subjects will be asked to answer the question “I chose the idea that I thought was the most creative” (1 = completely disagree–7 = completely agree). For the How and Why? startup task manipulation, Study 2 drew on previous research ([35]) by asking two raters to rate the level of construal based on subjects’ responses to why or how.

The subjects’ responses were coded as −1 if they described a lower level than “maintain a good mood”, +1 if they described a higher level than “maintain a good mood”, and 0 if they did not meet either criterion. The ratings of the subjects’ four responses were summed to form an indicator of the level of construal. The average of the two raters’ scores was taken as the subject’s final score. The manipulation was considered valid if the Why group score was higher than the group mean or the How group score was lower than the group mean.

Apparatus

The equipment for this study included desktop computers and electronic files. All computers had partitions between them, and the mouse and keyboard were silent devices to ensure that the subjects’ research environment was relatively independent, thus eliminating social comparison or convenience interference.

#### 4.1.3. Procedure

Teenagers completed this research in a quiet computer room with the research procedure shown in Figure 6 below.

The researcher explained the research procedure to the subjects, including the requirements of state construal-level tasks, the criteria for selecting creative ideas (as in Study 1), and the manipulation test requirements. This explanation lasted 3 min.

Afterwards, subjects randomly assigned to low state-level construals were asked to think about and answer the question “How to maintain a good mood”, while those with high state-level construals were asked to think about and answer the question “Why to maintain a good mood”.

The adolescent subjects then completed the idea selection tasks for the general creation domain task (AUT) and the social creation domain task (emotional distress) as in Study 1. This phase lasted 10 min. Finally, the manipulation test was completed, also lasting 10 min.

We used Excel software for initial data organization and SPSS 27.0 for ANOVA and other analyses.

Regrettably, the results of the manipulation check were lost during the experiment, preventing further statistical analysis. However, the method is based on established research ([35]), and the manipulation effects were indirectly supported by the participants’ response patterns.

### 4.2. Study 2 Results

In Study 2, the performance of adolescents with high and low levels of state construal in the general creative domain and social creative domain tasks is presented in Table 2. To test the interaction effect of state-level construals and task domain on adolescents’ creative idea selection, three 2 (high level of state construal, low level of state construal) × 2 (general creativity domain task, social creativity domain task) repeated-measures ANOVAs were conducted. Creativity, novelty, and applicability were the dependent variables, state-level construals were the between-group variable, and task domain was the within-group variable.

For creativity, the main effect of state-level construals was significant, *F* (1, 127) = 13.18, *p* < 0.001, *η*^2^*_p_* = 0.09. Creativity scores for idea selection were significantly higher in subjects with high construal levels than those with low construal levels. The main effect of the task domain was significant, *F* (1, 127) = 20.49, *p* < 0.001, *η*^2^*_p_* = 0.14. Creativity scores for idea selection were significantly higher in the general creativity domain task than in the social creativity domain task. The interaction effect between state-level construals and task domain was not significant, *F* (1, 127) = 0.06, *p* > 0.05, *η*^2^*_p_* = 0.000.

For novelty, the main effect of state-level construals was not significant, *F* (1, 127) = 0.24, *p* > 0.05, *η*^2^*_p_* = 0.002. The main effect of the task domain was significant, *F* (1, 127) = 43.01, *p* < 0.001, *η*^2^*_p_* = 0.25. Novelty scores for idea selection were significantly higher in the general creativity domain task than in the social creativity domain task. The interaction effect between state-level construals and task domain was not significant, *F* (1, 127) = 0.51, *p* > 0.05, *η*^2^*_p_* = 0.004.

For applicability, the main effect of state-level construals was significant, *F* (1, 127) = 6.14, *p* < 0.05, *η*^2^*_p_* = 0.05. Subjects with high levels of construal scored significantly higher in terms of applicability than those with low levels of construal. The main effect of the task domain was not significant, *F* (1, 127) = 0.07, *p* > 0.05, *η*^2^*_p_* = 0.001. The interaction effect of state-level construals with the task domain was also not significant, *F* (1, 127) = 0.003, *p* > 0.05, *η*^2^*_p_* = 0.000.

According to the results of Study 2, there was a significant main effect of state-level construals on the creativity and applicability of adolescents’ creative idea selection. Adolescents with high levels of state construal scored significantly higher on creativity and applicability than those with low levels of state construal in both general and social creativity domain tasks. This indicates that adolescents with high levels of state construal showed higher creativity and applicability in the research, validating hypothesis H2a.

The main effect of the task domain on creativity and novelty is significant. The scores of adolescents’ creative idea selection in general creative domain tasks are significantly higher than those in social creative domain tasks, which supports the task domain differences proposed in hypothesis H2b. However, the interaction effect between state-level construals and task domains is not significant, indicating that high state-level construal has a similar impact on both task domains. Therefore, H2a is verified, and H2b is only partially supported.

### 4.3. Study 2 Discussion

To determine the consistency of trait and state levels of construal on adolescents’ social creativity idea selection, Study 2 compared differences in performance between high and low state levels of construal in general and social creativity domains.

#### 4.3.1. The Effect of State-Level Construals on Creative Idea Selection

The results of Study 2 indicate that adolescents with high state-level construals perform significantly better in the creativity of idea selection compared to those with low state-level construals. This finding is consistent with the results of Study 1 and supports the hypothesis that high state-level construals prompt adolescents to focus more on abstract, long-term, and goal-related information. However, in terms of the novelty of idea selection, Study 2 did not observe a significant advantage for adolescents with high state-level construals over those with low state-level construals. Notably, Study 2 found that adolescents with high state-level construals performed significantly better in terms of applicability compared to their counterparts with low state-level construals. This result not only differs from Study 1 but also challenges the traditional notion that high-level construals are generally more inclined towards abstract thinking.

Changes in state-level construals can drive individuals to switch between abstract and concrete cognition, and their impact is more dynamic and task-dependent. The results of Study 2 show that in familiar task contexts, adolescents with high state-level construals perform significantly better in applicability assessments compared to those with low state-level construals. This indicates that high state-level construals enable adolescents to flexibly adjust their cognitive strategies based on specific task requirements, making practical feasibility a key consideration in their creative idea selection. This dynamic adaptability stems from the sensitivity of state-level construals to task details ([54], [55]), allowing adolescents to more accurately assess the potential value of ideas in practice. However, this tendency is not advantageous in all contexts. When tasks emphasize the novelty or breakthrough of creativity, excessive focus on applicability may limit adolescents’ creativity.

Additionally, there may be a dynamic trade-off relationship between applicability and novelty. Study 2 found that when adolescents focus more on the practical value and feasibility of ideas, they may reduce their pursuit of novelty or uniqueness. This balance suggests that applicability and novelty, as independent dimensions of creative idea selection, may show differentiated importance in different contexts. As task familiarity increases, adolescents with high state-level construals tend to focus on more practical and concrete evaluation criteria. The accumulation of experience prompts their thinking to shift from abstract, long-term goals to more realistic and feasible specific goals ([15]). This cognitive process not only reveals the dynamic impact of state-level construals on the evaluation of different dimensions of creative selection but also highlights the critical role of task context and cognitive strategy adjustments in creative idea selection.

Thus, the role of state-level construals is not only reflected in the advantages of applicability assessment but also involves how task demands guide adolescents in making appropriate dynamic adjustments between applicability and novelty. When designing and evaluating practical tasks, it is essential to consider how context shapes cognitive strategies. Through reasonable guidance, we can help adolescents achieve a balance between different dimensions, thereby enhancing the overall quality of creative idea selection.

#### 4.3.2. The Effect of Task Domain on Creative Idea Selection

Study 2 showed that the task domain had a significant effect on the creativity and novelty of adolescents’ idea selection. Adolescents’ idea selection scored significantly higher on creativity and novelty in the general creativity domain task than in the social creativity domain task. This finding is not only consistent with Study 1 but also validates the findings of existing studies. [23] ([23]) pointed out that the general creativity task focuses more on the novelty of ideas, while the social creativity task emphasizes the balance between novelty and applicability. Although tasks in the general creativity domain were more likely to stimulate adolescents’ creativity and novelty, underperformance in the social creativity domain remained significant. This may be related to the specificity of social creativity tasks, which usually require consideration of more realistic contextual and social factors, thus requiring adolescents to balance between creativity and practical applicability.

However, the deficiencies of adolescents in the social creative domain may reflect the challenges they face in evaluating such tasks. This is especially true because these tasks require not only an accurate judgment of creativity and novelty but also strong social cognition, situation judgment, and collaborative abilities ([22]). The development of these abilities usually takes time and may be restricted by adolescents’ experience and knowledge accumulation. Therefore, training in social creative assessment tasks is particularly important. It not only helps adolescents find a balance between creativity and applicability but also cultivates their social responsibility and critical thinking, supporting their all-round development.

#### 4.3.3. The Effects of State-Level Construals and Task Domains on Creative Idea Selection

Study 2 did not find an interaction effect between state-level construals and task domain. The creativity, applicability, and novelty effects of high levels of state construal on adolescents’ idea selection showed similar trends in general and social creativity domain tasks. This result is consistent with Study 1, suggesting that the effects of state-level construals did not change significantly by the task domain. Nonetheless, the overall strengths of adolescents with high levels of state construal further validate their facilitating effect on creative idea selection and highlight the research value and practical implications of enhancing adolescents’ social creativity performance.

## 5. Conclusions and Implications

Combining the results of the two studies, adolescents’ accuracy in selecting the most creative and novel ideas in general creative domain tasks is significantly higher than that in social creative domain tasks. Adolescents with high trait- and state-level construals show higher accuracy in selecting creative ideas, and adolescents with high state-level construals also show an advantage in selecting more applicable ideas.

Firstly, the task domain significantly influences creative idea selection performance, particularly in general creative tasks where adolescents exhibit higher accuracy in selecting the most creative and novel ideas. This domain difference not only deepens the understanding of the creative selection process but also provides practical directions for refining educational strategies. This study indicates that tasks in the social creative domain often involve more practical applications and social relevance requirements, which may pose challenges for adolescents in selecting novel ideas. Therefore, teachers can tailor their teaching content based on task characteristics and design more challenging and open-ended creative tasks, such as “creative proposals for future technology products”, with a particular focus on stimulating students’ creative thinking in general creative tasks. To address adolescents’ deficiencies in social creative tasks, teachers can enhance the practical applicability and social relevance of the tasks, helping students to find a balance between novelty and applicability. For instance, they can design creative tasks based on community issues in the form of group projects (e.g., “community environmental solutions”) to improve their performance in creative idea selection for social creative tasks.

Secondly, trait- and state-level construals have a significant impact on adolescents’ creative idea selection performance. This study verifies the robustness and consistency of these two level construals, further enriching the theoretical connotation of level construals. Educators should adopt differentiated teaching strategies for students with construals of different levels. For instance, adolescents with higher state-level construals benefit from more abstract and flexible tasks that promote independent thinking and creative potential. Simultaneously, incorporating collaboration and interaction can help students with high state-level construals combine creativity with practical applications, especially in social creative domain tasks. These personalized teaching strategies can better stimulate students’ creative potential, enhancing their performance across different task domains, thus providing robust support for the overall development of adolescents.

By revealing task domain differences, verifying the impact mechanisms of level construals on creative idea selection performance, and proposing personalized educational strategies, this study not only fills the theoretical gap in the field of adolescent creative idea selection but also provides a scientific basis for educational practice. Specifically, it offers insights into how to optimize teaching design based on different task domains and level construals to enhance adolescents’ creative idea selection performance.

## 6. Limitations and Future Directions

This study deepens the research on the creative process based on the Creativity Stage Theory at the behavioral level ([41]), enriches the research on general and specific domains in the field of creativity, and provides useful empirical evidence to enhance adolescents’ creativity. However, the findings reveal several limitations that need improvement and refinement in future studies.

First, this study revealed that high levels of construal had a significant advantage over low levels in adolescents’ creative idea selection. However, some results did not align with expectations, suggesting that the effects of high and low levels of construal may not be constant but may be moderated by unclarified variables. Future research can explore these potential moderating variables to further clarify the mechanisms of different construal levels on creative idea selection.

Second, this study mainly demonstrated the positive effects of high levels of construal, potentially causing low levels of construal to be neglected or produce negative value judgments. It should be emphasized that the construal level is not absolutely superior or inferior; its effect largely depends on the needs of the specific context and task goals. When a high level of construal matches the target’s needs, its effect may be more significant; in other cases, a low level of construal may be more adaptive. Future research should remain open and cautious, exploring the unique value of both construal levels in different contexts, to provide more comprehensive theoretical perspectives and practical guidance.

Additionally, there are inherent differences in complexity between the general and social creative tasks in this study, which may affect participants’ performance and the generalizability of the results. Although we controlled for these differences through methods such as the random assignment of task conditions and standardized scoring, future research should further balance task complexity and explore the role of task difficulty as an independent variable while also considering the potential impact of task order.

## Figures and Tables

**Figure 1 jintelligence-13-00031-f001:**
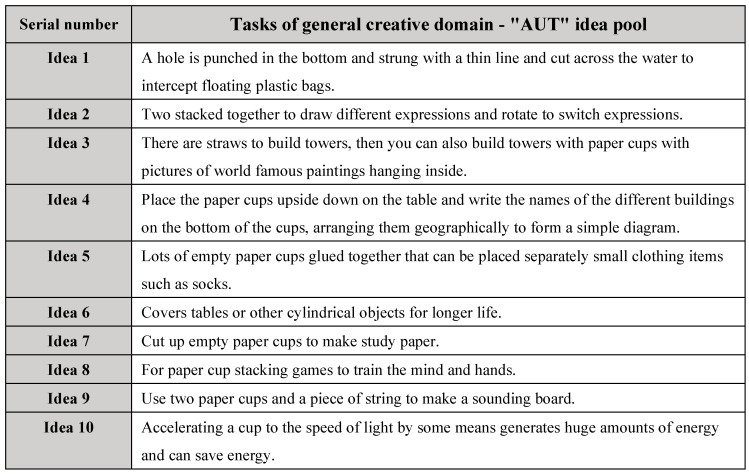
Tasks of general creative domain—“AUT” idea pool.

**Figure 2 jintelligence-13-00031-f002:**
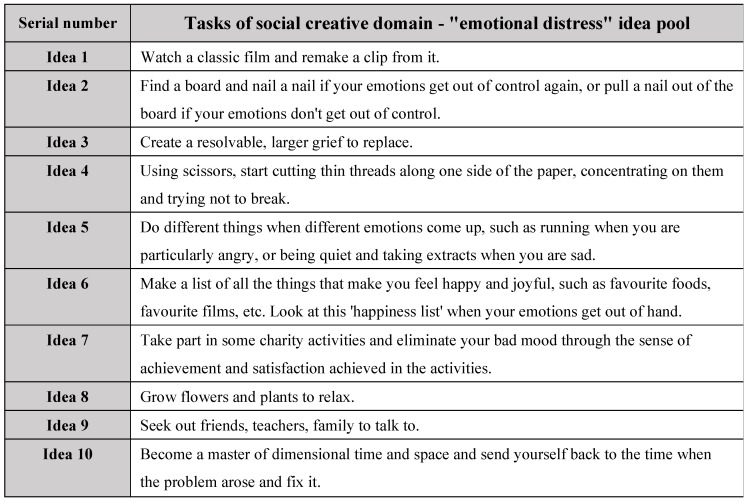
Tasks of social creative domain—“emotional distress” idea pool.

**Figure 3 jintelligence-13-00031-f003:**
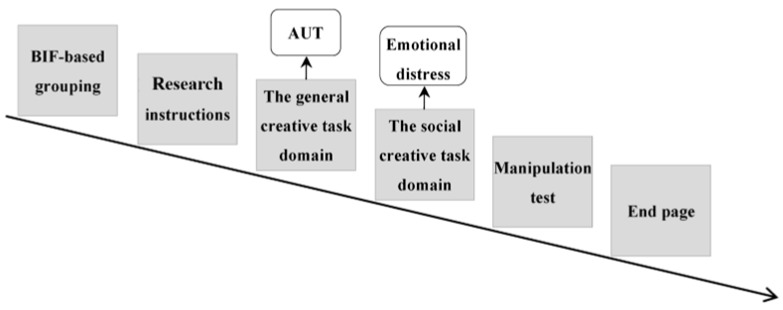
Effects of trait-level construals on adolescents’ creative idea selection: a comparative research procedure based on task domains.

**Figure 4 jintelligence-13-00031-f004:**
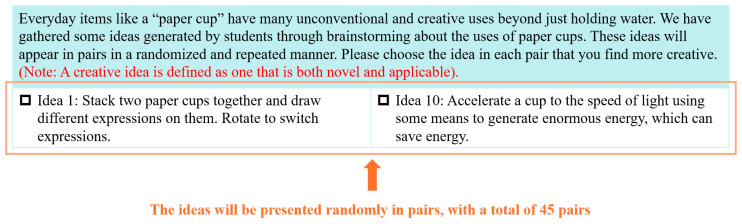
The interface of idea selection task–AUT task as an example.

**Figure 5 jintelligence-13-00031-f005:**
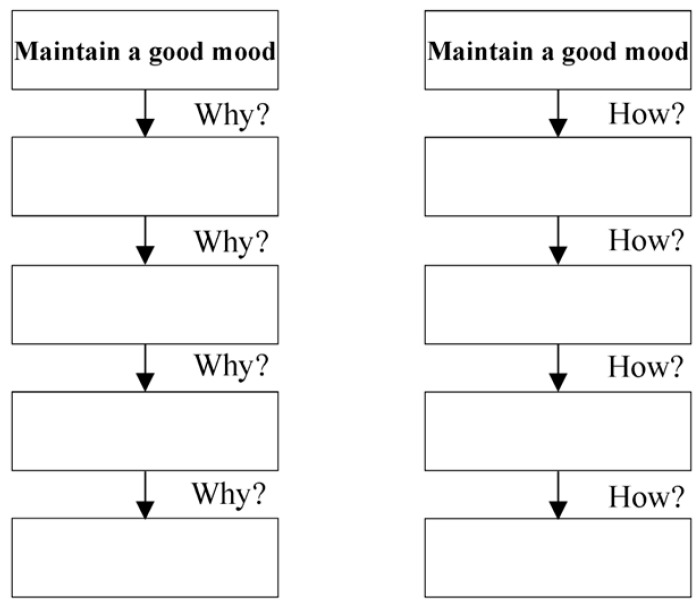
Schematic diagram of How and Why? startup task.

**Figure 6 jintelligence-13-00031-f006:**
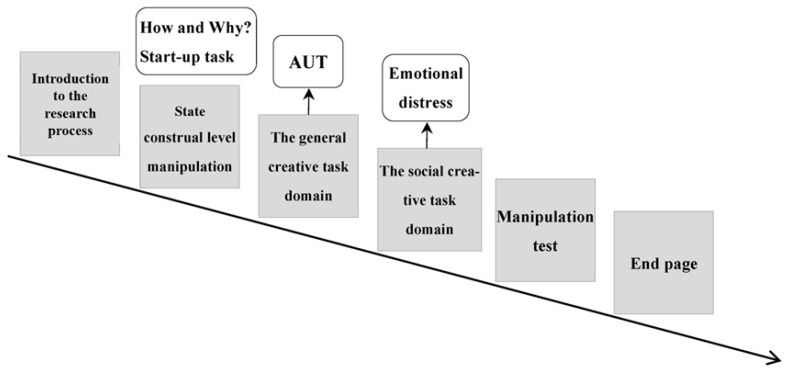
Effects of state-level construals on adolescents’ creative idea selection: a comparative research procedure based on task domains.

**Table 1 jintelligence-13-00031-t001:** The creative idea selection performance of participants with different trait-level construals on different domains of tasks.

BIF	Task Domain	Aspects of Creative Idea Selection		
		Creativity	Novelty	Applicability
High ^a^	General	3.60 (1.49)	4.22 (1.04)	4.34 (1.45)
	Social	3.00 (1.32)	3.31 (1.29)	4.32 (1.20)
Low ^b^	General	2.89 (1.51)	3.94 (1.18)	3.85 (1.73)
	Social	2.77 (1.48)	3.20 (1.48)	4.29 (1.23)

Note: ^a^ n = 65; ^b^ n = 65.

**Table 2 jintelligence-13-00031-t002:** The creative idea selection performance of participants with different state-level construals on different domains of tasks.

BIF	Task Domain	Aspects of Creative Idea Selection		
		Creativity	Novelty	Applicability
High ^a^	General	3.71 (1.43)	4.14 (1.12)	4.51 (1.25)
	Social	2.92 (1.33)	3.11 (1.32)	4.55 (0.95)
Low ^b^	General	3.02 (1.55)	3.95 (1.21)	4.02 (1.70)
	Social	2.31 (1.19)	3.13 (1.37)	4.05 (1.65)

Note: ^a^ n = 65, ^b^ n = 64.

## Data Availability

The raw data supporting the conclusions of this article will be made available by the authors on request.

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
