# Peer review of "The Role of Abstraction: Construal Level Can Improve Adolescents’ Idea Selection in the Domain of Social Creativity"

_jintelligence, 2025, doi:10.3390/jintelligence13030031_

Round 1
Reviewer 1 Report
Comments and Suggestions for Authors
This is an interesting paper that addresses a little studied topic. There are several issues that could help improve the work.
H3a: why is an interaction hypothesized, how does it work ?
Experiments; what is the power of the studies given the sample size and expected effects?
Participants:; what is the country origin of participants, were they tested at school ?
BIF test, it seems odd to select a random subset of questions.
Social creative task: it is not clear if participants had an instruction to be creative , compared to the AUT task. This would be an important confound . There is also an order of task issue. So, the social task seems to come always after the AUT task. It is therefore hard to say if the social domain has a lower creativity score because it is a social domain, or due to the order effect ( more critical in a second task), or due to a lack of a "be creative" instruction. therefore all results for the social task seem hard to interprete.
The interpretation to explain why there is a lack of construal result for novelty is not convincing. So the two studies in the arrticle show contradictory results, and this is quite problematic.
Comments on the Quality of English Language
The english can be improved. some formulations and sentences seem awkward or not aligned with standard english.
Author Response
Thank you for your comments concerning our manuscript. According to them, we revised the manuscript, and the responses to the comments are as following. Please see the attachment.

Reviewer 2 Report
Comments and Suggestions for Authors
The current manuscript details one correlational and one experimental study on the relation between creativity and construal level. Idea selection and applicability were higher when construal levels were high, but there was no effect on novelty. This is a small, but potentially important contribution to the literature. Unfortunately, at this time, I cannot recommend this article for publication because of major issues with the presentation, explanation, and interpretation of the studies. I detail some of these issues below, roughly in the order that they appear in the manuscript.
1. The authors refer to chronic and situational construal levels. Typically, in psychology these would be referred to as trait- and state-level construals, respectively.
2. Study 1 is not an experiment and thus should not be referred to as such.
3. More detail is needed in the introduction to support the authors’ arguments. The authors should focus on providing clearer connections between concepts, fully supporting their hypotheses with specific empirical and/or theoretical evidence, making more convincing arguments. For example, around line 50, the authors state: “Given the close association between creative thinking, the prefrontal cortex, and cognitive control function, adolescence constitutes a malleable stage in which the development of creative thinking can take place.” Further elaboration is required to explain these connections; this statement of correlations is not a sufficient explanation. I will underscore that this is just one example. The authors should carefully examine the whole paper to make sure that their arguments are explained well. In-text citations are necessary, but not sufficient.
4. On the other hand, in several places, the authors unnecessarily repeat themselves. For example, much of the content under the “2. Experiment 1” header was already covered in the general introduction.
5. In a few places, the authors mention psychological distance but do not really delve into the related literature on this topic at all. It could be helpful in supporting their hypotheses and I recommend they look into it further.
6. Support H3a and b interaction hypotheses. What exactly do you expect and why?
7. Please provide more discussion of the similarities and differences between the AUT and emotional distress tasks. Given the descriptions provided in lines 198-202, the tests seem to have varying levels of difficulty in addition to the differences in topics covered. Asking if there are other uses for a cup leaves open any number of possibilities, whereas Nonan’s example is relatively constrained and what he *should* do would naturally have fewer acceptable responses. To what extent do you think that these different task demands influence the results of this study (i.e., that scores were higher for general creativity than social).
8. On that note, if the task domain results are valid, the authors should provide more discussion of the developmental reasons as to why adolescents might show these differences.
9. Data in tables should be presented as bar graphs.
10. Provide exact p values.
11. Experiment 1 Discussion is again repetitive.
12. The authors are overly confident in their interpretations and should temper the language used. For example, (in line 330) “*Obviously*, their intention to choose.” This does not seem obvious at all, and there could be other explanations. In line 556, they mention that “the urgency is self-evident...” That might be true for the authors, but not for the reader. This needs additional explanation and possibly toning down.
13. Line 340: The authors mention that adolescence is a “critical period for the development of creative identity.” Critical period is a technical term and not what I think the authors mean to say here. Do you meant to simply referring to this as important? If not, you’ll need more evidence to support the idea that this is, in fact, a “critical” period.
14. Provide age ranges for participants.
15. Coding explanation is not sufficient. What do you mean by upper and lower levels? Another researcher should be able to replicate this.
16. Provide statistics for manipulation check.
17. Are the AUT and emotional distress questions counterbalanced?
18. Section 3.2: Cut interpretation of descriptive statistics.
19. Be consistent with labels for experimental conditions. At one point you refer to them as High v. Low situational construal, at another as high-explanatory level.
20. The authors hypothesize that “over time… the familiarity of the adolescent participants with the ideas would have gradually increased.” (line 591) Do you have the data to confirm this hypothesis?
21. Final discussion is insufficient in terms of wrapping up the importance of this research and situating it in the broader literature.
Comments on the Quality of English LanguageThe quality of English in this manuscript is OK, but clarity is an issue throughout. In addition to what I've covered above, the authors should pay close attention to word use, the length of sentences, and agreement between clauses. I've noted a few instances that stuck out to me below, but I encourage the authors to work with an editor.
Line 106, 348 – sentences are very unclear
Line 573: typo
Line 619 – “Modulators.” Moderators?
Author Response

(The authors gave the same response as above.)

Reviewer 3 Report
Comments and Suggestions for Authors
This is a very original article that explores whether idea selection in the field of social creativity can be improved by the level of construal achieved.
The article truly addresses a little-studied aspect of creativity, which is creative idea selection, and tries to provide some characteristics of the process.
The work is presented in a very complex manner with appropriate evaluation instruments. The two experiments are well designed. The results are somewhat paradoxical, but they are correct.
I understand that the conclusions highlight the relevance of the general domain of creativity, which makes them of interest.
Author Response

(The authors gave the same response as above.)

Round 2
Reviewer 1 Report
Comments and Suggestions for Authors
This revised paper shows good progress. There are however some additional points.
1. In study 1, it is noted tahta t test and anova are done. Howe er, there is no t -test.
2. the performance on the social creativity task is usually less good than the AUT task. This may be due to the more restriucted range of creative idaes ( less range in high to low novelty) , which makes the judging task harder. Otherwise, as the social judgment task comes after the AUT task, people may be more tired, less attentive, and do less well.
3. The AUT task , object use , is called "general" creativity. However, it is better to see it as "object-related" creativity, and that contrasts with "social situation" creativity
4. It is possible that the high construal subjects, think more abstractly, but also are more intelligent in an IQ type way. This general intellige,ce would also favor evaluating the creativity of idaes with greater accuracy. This alternative interpretation can be mentionned.
5. It is stated that H1b in study 1, and H2 in study 2is partly supported. however, This seems not to be the case.
6. The results show that creativity judgments are more accurate for high construal subjects and for the object task . The subjects must select the most creative/ novel ideas. They do not generate ideas themselves. However, the way the results are described, it sometimes reads that subjects with high construal or doing the object task were ** more creative**. This is a misinterpretation, and needs to be fixed. see lines: 318-321, 554-559, 536-543).
7. the logic of lines 536-543 is not clear.
8. line 373_375 tasks did not perform better... noeedds to be reworked
9; line 392 what is "re-randomly "recruited ?
Comments on the Quality of English LanguageThere are some sentences that seem odd, not reflecting standard english. Some sentences need to be improved.
Author Response
Thank you again for your valuable comments on our paper. According to them, we revised the manuscript, and the responses to the comments are as following. Please see the attachment.

Reviewer 2 Report
Comments and Suggestions for Authors
Overall, I found this manuscript to be much improved. However, the following issues remain:
1. The authors took great care to clarify the introduction. However, in doing so, some aspects of the background were trimmed too much. For example, in lines 96-97, they talk about the weights used in decision-making, which would not be clear to a non-expert reader without further elaboration. Also, after revision, there is still some repetition of concepts: e.g., lines 129-131 and lines 138-140.
2. Additional citations are needed throughout the introduction to support claims being made. Also, in lines 121-123, the example comes from a paper by Lieberman & Trope, but is not cited as such. Similarly, the discussion misses citations (e.g., line 582 for Creativity Stage Theory) and does not reference other outside work; this is absolutely necessary to support the authors' claims and situate these studies within the broader literature.
3. The authors pay very little attention to the population used in this study. The fact that participants are adolescents is treated as an afterthought. But, this is an important consideration for the study that merits two additions to the manuscript: 1) significant discussion of the developmental stage and how it could impact creativity, construal, and/or social context, and 2) statistical tests to determine whether age a) affects the results in any way and b) is equally distributed across experimental conditions in study 2.
4. In study 1, would the results hold if the results were not dichotomized?
5. One of the biggest issues for this manuscript has not been properly addressed in the manuscript since the last review: the differences in difficulty between the general and social creativity tasks. Although the authors mention the difficulty levels, they do not sufficiently address how they could affect the results and how they could limit the contributions of this work.
6. In study 2: 1) Do you have any checks that could help to ensure random assignment to condition? BIF, for example? Also, age? 2) Interrater reliability?
7. The authors mention a manipulation check for Study 2, but do nothing with that information. It is unfortunate that the data were lost. That should be clarified in the manuscript.
8. Section 4.3.1: Expand and clarify discussion of applicability. Needs better theoretical rationale.
9. Section 5: There is no discussion of how this study contributes to the broader literature. The authors should wrap up the take-home messages of both studies in this section. Also, the recommendations for educators feel like they come out of nowhere and much of the section is a stretch given the current studies. The authors should, at the very least, combine their own research with other research from the field to properly support the claims being made.
Minor questions:
1. Line 183: What is a view?
2. Line 193: what were the ideas selected for?
3. Lines 203-206: Relevance to this study?
4. Figure 3: still says "experimental instructions"
5. Line 457: language implies that you've done a manipulation check to ensure construals were different across conditions.
Comments on the Quality of English Language
The quality of English is good, especially in the introduction. However, there are several issues related to word choice (e.g., line 71 "emerged") and multiple typos throughout the paper (e.g., lines 34-35 "is" should be "if")
Author Response

(The authors gave the same response as above.)

Round 3
Reviewer 1 Report
Comments and Suggestions for Authors
Overall suggestion that it can be accepted after minor revisions;
Remaining issues:
The English expressions need to be fixed in some further spots.
The section 1.1 on domain specificity research , and the part of research on social creativity as a domain needs to be updated. It is missing recent work.
Hypothesis 1b in study 1 is not well justified. Why should the performance be better in the general task ?
The median split in study 1, on trait construal is at 8 points , on a scale of 12 to 10. This suggests that the low construal group is not so low. It should be discussed.
Section 3.3.3 is not clear, and probably should be removed. It seems to interpret non significant results.
The significance tests may be "two-sided" , non directional. However, if the hypothesis is directional, perhaps a one sided P can be used;, and a few results may then be significant.
Comments on the Quality of English Languageno comment
Author Response
Thank you for your comments concerning our manuscript. According to them, we revised the manuscript, and the responses to the comments are as following.